# Encapsulation of Rosemary Extracts using High Voltage Electrical Discharge in Calcium Alginate/Zein/Hydroxypropyl Methylcellulose Microparticles

**DOI:** 10.3390/foods12081570

**Published:** 2023-04-07

**Authors:** Marinela Nutrizio, Slaven Jurić, Damir Kucljak, Silvija Lea Švaljek, Kristina Vlahoviček-Kahlina, Anet Režek Jambrak, Marko Vinceković

**Affiliations:** 1Faculty of Food Technology and Biotechnology, University of Zagreb, 10000 Zagreb, Croatia; ddamirkucljak@gmail.com (D.K.); silvija.lea1710@gmail.com (S.L.Š.); anet.rezek.jambrak@pbf.unizg.hr (A.R.J.); 2Faculty of Agriculture, University of Zagreb, 10000 Zagreb, Croatia; sjuric@agr.hr (S.J.); kvkahlina@agr.hr (K.V.-K.); mvincekovic@agr.hr (M.V.)

**Keywords:** microencapsulation, rosemary extract, high-voltage electrical discharge, bioactive compounds, alginate, zein, hydroxypropyl methylcellulose

## Abstract

The increased demand for functional food with added health benefits is directing industrial procedures toward more sustainable production of naturally added bioactive compounds. The objective of this research was to investigate the potential of bioactive compounds from rosemary extract obtained using high-voltage electrical discharge as a green extraction method, for microencapsulation as a protective method for future application in functional food. Four types of microparticles were made via the ionic gelation method using alginate (Alg), zein (Z), and hydroxypropyl methylcellulose (HPMC) biopolymers and were analyzed considering the physicochemical properties. The diameter of dry microparticles ranged from 651.29 to 1087.37 μm. The shape and morphology analysis of microparticles showed that the obtained microparticles were quite spherical with a granular surface. The high encapsulation efficiency was obtained with a loading capacity of polyphenols up to 11.31 ± 1.47 mg GAE/g (Alg/Z microparticles). The microencapsulation method showed protective effects for rosemary polyphenols against pH changes during digestion. Specifically, the addition of both zein and HPMC to calcium-alginate resulted in microparticles with a prolonged release for better availability of polyphenols in the intestine. This research background indicates that the release of rosemary extract is highly dependent on the initial biopolymer composition with high potential for further functional food applications.

## 1. Introduction

Consumer interest in the beneficial health effects of natural bioactive compounds with antioxidant activity is continuously growing. This is especially noticeable in the food, pharmaceutical, and cosmetic industries, with an emphasis on bioactive compound bioavailability, metabolism, and biological effects [1,2]. Nevertheless, antioxidants, particularly polyphenols, are not stable and they easily interact with other compounds from matrices that surround them. For that reason, it is important to find the best methods for their management from the extraction process to the final incorporation into a new product [3,4].

Firstly, a suitable extraction method should be considered. Due to the disadvantages of conventional extraction techniques that include long processing times and high working temperatures, innovative non-thermal extraction techniques present a favorable alternative for the extraction of bioactive compounds [5,6]. In 2012, Chemat et al. introduced the concept of green extraction of natural products that are within the principles of green chemistry and engineering [7]. Such an extraction method is a high-voltage electrical discharge (HVED) that is a novel, efficient, eco-friendly extraction method which, in comparison with conventional methods, works with reduced solvent consumption, low operating temperatures, and temperature rise during the extraction, higher extraction yield, and less processing time [8,9]. The HVED principle of operation is based on electroporation [9,10], a phenomenon caused by the electrical field that increases the permeability of the cell membrane, allowing intracellular bioactive compounds to easily diffuse into the extraction solvent [11].

Due to its structure and nature, the aqueous extract reached in polyphenols is not a stable medium for further processing. Their unsaturated bonds in the molecular structure make them sensitive to external environmental conditions such as oxidants, light, pH and temperature changes, enzymatic activities, etc. [1,4]. Therefore, the polyphenolic stability should be increased via protection from external conditions for safe delivery. Microencapsulation is an emerging technology that ensures the protection of sensitive compounds against various processing conditions by encapsulating them inside a coating material and providing a controlled release under specific conditions. Additionally, the food safety and sensory quality of the product can also be improved [12]. Depending on their rheological and functional properties and intended use, various coatings can be used for microencapsulation, either polymeric or nonpolymeric materials. Some coating materials include carbohydrates (such as starch, dextran, cellulose, alginate, pectin, carrageenan), proteins (such as gluten, casein, gelatin, zein), and others (such as glycol, polyethylene, cellulose derivatives) [13,14]. Zein is a class of prolamin proteins obtained from corn. It is composed of approximately equal amounts of hydrophilic and hydrophobic amino acid residues, and these amphiphilic properties ensure a high potential for zein to form microstructures such as spheres and films. Therefore, this edible coating material presents a good option for the microencapsulation of functional ingredients such as polyphenols and essential oils due to its biocompatibility, low water uptake value, thermal resistance, and excellent mechanical properties [15,16,17]. However, zein has been continuously reported as an encapsulating material for hydrophobic compounds, while studies are lacking regarding its usage for hydrophilic substances [18]. Hydroxypropyl methylcellulose (HPMC) is a partly O-methylated and O-(2-hydroxypropylated) cellulose ether derivative that is odorless, tasteless, and non-toxic. It is widely used in oral controlled delivery systems and can be used as a matrix for both hydrophilic and hydrophobic constituents [19,20]. The use of biodegradable polysaccharides, combining sodium alginate with HPMC, has been presented as a suitable combination for controlled drug delivery systems including bioactive compounds [21].

Rosemary (*Rosmarinus officinalis* L.) is a long-lasting evergreen aromatic herb from the *Lamiaceae* family, typical of the Mediterranean region [22]. Rosemary leaves have traditionally been used in Mediterranean cuisine for improving the flavor of food, for food preservation, and as a medicinal herb for its astringent, anti-inflammatory, antioxidant, antimicrobial, antiaging, antirheumatic, analgesic, and hypotensive properties [23,24]. The beneficial properties of rosemary are mostly related to its bioactive compounds, specifically phenolic compounds and flavonoids rosmarinic acid, carnosic acid, carnosol, and rosmanol [25]. Due to its therapeutic effects, rosemary extract, rich in bioactive compounds, presents a desirable natural substrate for encapsulation to preserve its properties for further production. Until today, rosemary essential oil was mostly used for encapsulation [26,27]. Only a few research papers reported the encapsulation of aqueous rosemary extracts and showed high potential for several applications in food technology or nanomedicine [28,29].

This study aimed to encapsulate rosemary extract obtained using HVED in biopolymer-based microparticles prepared from different combinations of calcium alginate, zein, and HPMC. The obtained microparticles were evaluated considering the physical properties including diameter, swelling degree, and morphology. Chemical analysis of the composition and its functional properties including encapsulation efficiency, loading capacity, and in vitro release profiles of polyphenols in simulated gastrointestinal tract conditions were also investigated. Additionally, a detailed characterization of molecular interactions between bioactive ingredients from the rosemary extract and encapsulating materials was performed. The results of this research will increase the possibility to use stabilized aqueous rosemary extracts obtained using green technologies in functional food preparation and the pharmaceutic/cosmetic industry.

## 2. Materials and Methods

### 2.1. Materials

Dried rosemary leaves (*Rosmarinus officinalis* L.) were provided by a local specialized drugstore (Suban d.o.o., Samobor). Plant material was collected in 2017, in the northwestern part of Croatia. Leaves were air-dried and stored in polyethylene bags in a dark and dry place, at ambient temperature before extractions were performed. Dried rosemary leaves were ground to a plant particle size distribution of d(0.1) ≤ 39.683 μm; d(0.5) ≤ 224.816 μm; d(0.9) ≤ 425.819 μm measured using the laser particle size analyzer Mastersizer 2000 (Malvern Instruments GmbH, Herrenberg, Germany).

Alginic acid sodium salt (CAS Number: 9005-38-3, M/G ratio of ~1.56, molecular weight 280,000) g/mol was purchased from Sigma Aldrich (USA). A commercially available product CaCl_2_ was purchased from Gram-Mol (Croatia), zein from Acros Organics BVBA (Belgium), and hydroxypropyl methylcellulose (HPMC) from Alfa Aesar (Germany). All chemicals were of analytical grade and used as received without further purification.

### 2.2. Methods

#### 2.2.1. Extraction Using HVED

HVED extraction was conducted using the “IMP-SSPG-1200” generator (Impel group d.o.o., Zagreb, Croatia) that generated rectangular pulses using direct current and achieving high voltage. Based on the previous study by Nutrizio et al. [10], optimized HVED parameters were chosen for the extraction from rosemary: frequency of 100 Hz, high-voltage current of 30 mA, pulse width 0.4 µs, voltage 25 kV using nitrogen as a reaction gas, with the gap between electrodes of 15 mm, treatment duration 9 min, and ratio mass to solvent 1:50 (*w/v*) (according to pharmacopeia). The mixture of herb material and the solvent was transferred to a beaker shaped reactor of 100 mL. The reactor, which is opened on both sides, was fitted with silicone tops that were 1 cm in diameter. Silicone tops were used due to easier mounting of the electrode from the top and needle from the bottom. The setup of the generator and reactor, as well as a detailed extraction description, have been described previously [10].

#### 2.2.2. Microparticle Preparation

Microparticles were prepared via ionic gelation at room temperature in a sterile environment as described by Jurić et al. [30]. Ionic gelation involved the preparation of microparticles by dripping a solution of sodium alginate without or with zein and/or HPMC using Encapsulator Büchi-B390 (BÜCHI Labortechnik AG, Flawil, Switzerland) under constant magnetic stirring. The total concentration of coating materials was constant (1.5% *w/v*), and four types of microparticles were made: 1.5% sodium alginate (Alg), 1.3% sodium alginate + 0.2% zein (Alg/Z), 1.2% sodium alginate + 0.3% HPMC (Alg/HPMC), and 1.0% sodium alginate + 0.2% zein + 0.3% HPMC (Alg/Z/HPMC). Before mixing with other biopolymers, zein was dissolved in distilled water (100 g/L (*w/v*)) and 1 mL of 1 mol/L NaOH was added. The solution was stirred for 30 min at 50 °C. The mixtures were dripped into 2% CaCl_2_ (*w/v*) solution through the encapsulator nozzle size of 1000 μm. Both coating materials and CaCl_2_ were dissolved in the rosemary extract obtained using HVED. Encapsulation parameters are shown in Table 1.

To promote gel strengthening, formed microparticles were kept at room temperature for an additional 30 min under constant magnetic stirring. Afterward, microparticles were filtered, washed with deionized water, air-dried to constant weight, stored in plastic Falcon tubes at room temperature, and protected from light, until further studies.

For comparison, blank (without extract) microparticles were made with the same coating materials and the same encapsulation parameters, just using distilled water instead of the extract as loading material. All microparticle formulations were prepared in triplicates.

#### 2.2.3. Determination of Total Polyphenolic Content (TPC)

The Folin–Ciocalteu method was used to estimate the total polyphenol content (TPC), as reported by Jatoi et al. [31]. The data for the TPC of extracts were expressed as mg of gallic acid equivalent weight (GAE) per L of the aqueous supernatant or rosemary extract.

#### 2.2.4. Determination of Antioxidant Capacity of Extracts Using ABTS and DPPH Assays

2,20-Azino-bis(3-ethylbenzothiazoline-6-sulfonic acid) (ABTS) and 2,2-Diphenyl-1-picrylhydrazyl (DPPH) assays were used to determine the antioxidant capacity of rosemary extract. The Trolox equivalent antioxidant capacity was estimated using both the DPPH radical scavenging assay and the ABTS assay, as described previously by Vinceković et al. [32]. The results, obtained from triplicate analyses, were expressed as Trolox equivalents (TE) and derived from a calibration curve determined for Trolox (100–1000 μmol/L).

#### 2.2.5. Determination of Total Flavonoids of Extracts (TF)

The total flavonoid content (TF) was determined with a spectrophotometric method as previously described [33]. A total of 1 mL of extract was added to a 10 mL volumetric flask containing 4 mL of distilled water. A volume of 300 μL of NaNO_2_ (0.5 g/L (*w/v*)) solution was added to the suspension, and after 5 min, 300 μL of AlCl_3_ (1 g/L (*w/v*)) was added. After 6 min, 2 mL of NaOH (1 mol/L) was added to the mixture. The final volume was set to 10 mL with the addition of distilled water. Absorbance was measured at 360 nm and calculated as mg quercetin equivalents (QE) per L of extract (mg QE/L).

#### 2.2.6. Determination of Total Protein Content in Extracts (TP)

The total protein content (TP) was determined with the Lowry method using bovine serum albumin (BSA) as the standard protein [34]. The results were expressed as mg equivalent of BSA per volume (mL) of the aqueous supernatant or rosemary extract.

#### 2.2.7. Encapsulation Efficiency, Loading Capacity, and Swelling Degree of Microparticles

Detailed procedures for the determination of encapsulation efficiency (EE), loading capacity (LC), and swelling degree (S_w_) of microparticles were previously described [35,36].

EE expressed as a percentage was determined from the total TPC content (TPC*_tot_*) from polymer solution (TPC_p_), TPC content from CaCl_2_ solution (TPC_c_), and content in the filtrate (TPC_f_) and calculated using the equation:(1)$$EE(\%)=\frac{{TPC}_{load}}{{TPC}_{tot}}\times 100$$
where TPC*_tot_* = TPC_p_ + TPC_c_, and TPC*_load_* = TPC*_tot_* − TPC_f_.

LC expressed as a TPC content (mg GAE/g) was calculated with the equation:(2)$$LC=\frac{c\times V}{W_{c}}\times 100$$
where c is the concentration of TPC in the sample, V is the volume of the sample, and W_c_ is the weight of dry microparticles.

For S_w_ determination, dry microparticles were swelled in distilled water. Microparticle S_w_ was calculated using the equation:(3)$$S_{w}=\frac{w_{t}-w_{0}}{w_{0}}\times 100$$
where w_t_ is the weight of the swollen microparticles and w_0_ is their initial weight. All measurements were replicated three times, and the results are presented as the mean values.

#### 2.2.8. In Vitro Phenolic Release from Microparticles

In vitro release studies of TPC from microparticles were carried out at 37 °C in distilled water, and simulated gastric (HCl, pH 1.64) and intestinal solutions (pH 7.40, phosphate buffer −0.2 mol/L Na_2_HPO_4_, 0.2 mol/L NaH_2_PO_4_ × 2H_2_O). The release of polyphenolic compounds from the microparticles was performed by measuring the released TPC cumulative concentration using the Folin–Ciocalteu assay as described in Section 2.2.3. The results are presented as the percent of cumulatively released TPC (f) using the equation:(4)$$Cumulative\;polyphenolic\;release(\%)=\frac{{TPC}_{t}}{LC}\times 100$$
where TPC_t_ presents the cumulative concentration of TPC released in time t, and LC is the total amount of TPC loaded in microparticles.

#### 2.2.9. Fourier-Transform Infrared Spectroscopy Analysis

The Fourier-transform infrared spectroscopy (FTIR) spectra of individual constituents and microparticles were recorded with the FTIR Instrument—Cary 660 FTIR (MIR system) spectrometer (Agilent Technologies, Santa Clara, CA, USA). To make pellets, samples were combined with potassium bromide. The spectral scanning ranged from 500 to 4000 cm^−1^.

#### 2.2.10. Microscopic Observations

Microparticles were observed using three different microscopic techniques:
Optical microscopy (Leica MZ16a stereomicroscope, Leica Microsystems Ltd., Saint Gallen, Switzerland) was used to examine the size and shape of the microparticles. An average diameter of prepared dry microparticles was determined using Olympus Soft Imaging Solutions GmbH, version E_LCmicro_09Okt2009. Diameters of about 100 microparticles, randomly selected from batches produced in triplicate, were measured.Scanning electron microscopy (SEM) (FE-SEM, model JSM-7000 F, Jeol Ltd., Akishima City, Japan) was used to determine the microparticle morphology properties. Microparticles were put on the high-conductive graphite tape. Energy-dispersive X-ray spectroscopy (EDS) was used to determine the elemental composition of the surface. FE-SEM was linked to an EDS/INCA 350 (energy dispersive X-ray analyzer) manufactured by Oxford Instruments Ltd. (Abingdon, UK). Various compounds and elements were analyzed and marked as: C (CaCO_3_), O (SiO_2_), Na (Albite), Cl (KCl), Ca (CaCO_3_), Ca (CaCO_3_), Ca (CaCO_3_), Ca (CaCO_3_), Ca (CaCO_3_), Ca (C (Wollastonite)).The atomic force microscopy (AFM) using the MultiMode Scanning Probe Microscope with Nanoscope IIIa controller (Bruker Corporation, Billerica, MA, USA) was used to determine the surface morphology and obtain the topography of microparticles. The samples for AFM imaging were prepared by deposition of a microparticle suspension on the mica substrate. The microparticles were flushed three times with 50 μL MiliQ water to remove all residual impurities. The microparticle surface, cross-section, and grain size distribution within each sample were analyzed using MultiMode Scanning Probe Microscope with Nanoscope IIIa controller (Bruker Corporation, Billerica, MA, USA) with SJV-JV-130 V (“J” scanner with vertical engagement); Vertical engagement (JV) 125 μm scanner (Bruker Corporation, Billerica, MA, USA); Tapping mode silicon tips (R-TESPA, Bruker, Nom. Freq. 300 kHz, Nom. spring constant of 40 N/m). Accordingly, three-dimensional information about the surface topology was obtained and the roughness was quantified. All AFM imaging was performed at three different regions of each microparticle to ensure the consistency of obtained results.

#### 2.2.11. Statistical Analysis

All experiments were carried out in triplicates at room temperature. Results are presented as mean values ± standard deviation. Microsoft Excel 2016 and the XLSTAT statistical software add-in were used to examine the data.

## 3. Results and Discussion

The results and discussion are presented in three interconnected sections. In the first section, the chemical properties of obtained rosemary extracts are analyzed; the second section evaluates the physical properties of microparticles; and the final section discusses the functional properties of microparticles loaded with rosemary extracts.

### 3.1. Chemical Properties of Rosemary Extract for Encapsulation

The rosemary extract for encapsulation was obtained using the HVED method under optimal conditions determined in a previous paper by Nutrizio et al. [10]. For extraction purposes, distilled water was chosen as the cheapest and the most available green solvent. The rosemary extract was subjected to chemical methods to assess the composition of obtained extract for further encapsulation. Following the characterization of rosemary extract, the study of TPC, TF, TP, as well as antioxidant activity (ABTS and DPPH methods) was carried out. As shown in Table 2, the TPC of rosemary extract was 333.07 mg GAE/L which corresponds to 16.65 mg GAE/g of the dry weight of rosemary. This result is in line with previous results of the extraction of polyphenolic compounds from rosemary obtained using HVED [10]. In contrast, the TPC level of rosemary extract obtained with conventional extraction methods was significantly lower, as Alfonso et al. reported a TPC level in aqueous rosemary extract of 166.7 μg/g dry weight [37], and Pereira et al. measured a level of 409.1 μg/g dry weight in 80% ethanol extract [38].

Both antioxidant capacity methods, ABTS and DPPH, showed similar results: 1.94 mmol TE/L and 1.98 mmol TE/L, respectively. Antioxidant capacity in this work was also relatively higher when compared to that obtained via conventional extraction methods [1,28] and in line with extracts obtained with supercritical fluid extraction by Justo et al. [39]. The number of polyphenolic compounds and antioxidant capacity presented in this study were higher compared to other extraction methods, especially conventional ones. That may be related to the rosemary specifications such as environmental, harvesting, and genetic conditions, as well as different measurement methods, but mostly due to differences in the extraction procedure. The extraction using HVED induced electroporation of rosemary plant cells, which increases extraction potential and provides a higher extraction yield.

The TF in rosemary extract was 333.08 mg QE/L, which corresponds to 16.65 mg QE/L. The obtained result is in line with the research from Munekata et al. who found TF in rosemary extract obtained using various extraction methods, ranging from 16.29 (for ultrasound-assisted extraction) to 24.99 mg of catechin equivalents/g DW (for conventional extraction) [40].

### 3.2. Physical Properties of Microparticles Composed of Variable Coating Materials

In the second section, the physical properties of obtained microparticles are described to compare the morphological properties of microparticles prepared with rosemary extract to microparticles prepared with distilled water instead of the extract in the encapsulated core material. Furthermore, edible and readily available biopolymers were chosen as coating materials. Therefore, prepared microparticles were further analyzed to assess their properties and functionality. Physical properties included analysis of microparticle swelling behavior and diameter, as well as microscopic observations.

#### 3.2.1. Swelling Degree and Diameter of Microparticles

In Table 3, the swelling degree (%) and microparticle diameter (μm) are presented. The swelling behavior of prepared microparticles was analyzed in distilled water, as described in Section 2.2.7, while the diameter of microparticles was measured under the optical microscope (Figure 1).

When microparticles are dispersed in water, they swell. The swelling of microparticles depends on the hydrogel structure, temperature, properties, and composition of the core material [41]. To avoid the influence of electrolytes on swelling properties, the swelling behavior of prepared microparticles was observed in distilled water, and the results are shown in Table 3. Swelling degree (%) showed that there was a statistically significant difference for microparticles loaded with rosemary extract when HPMC was introduced in the structure. The swelling degree was 2.18–2.84 times higher with HPMC compared to microparticles prepared under the same conditions without HPMC. This is consistent with the results of Nochos et al., who found that swelling degree increases with higher HPMC content in microparticles [42]. The addition of zein did not significantly influence the swelling degree (*p* < 0.05). When compared to microparticles prepared with the same coating materials, but with distilled water instead of the rosemary extract, no significant difference was noted. Therefore, the addition of rosemary extract did not have a significant influence on the swelling behavior of microparticles. These results can be explained due to the high hydrophilicity and swellability of HPMC, as HPMC is the dominant hydrophilic polymer that swells significantly upon contact with water [42,43].

The prepared fresh microparticles were observed with the optical microscope and showed that microparticles were consistent in size (around 2 times larger than nozzle size) and had a spherical shape. However, after drying to constant mass, the regular spherical shape was lost and dents on the surface were noted (Figure 1). These results are consistent with the results of Jurić et al. where the surfaces of dry microparticles and wet microparticles were compared [44]. The wet microparticles kept their oval shape, unlike the dry microparticles, which had visible indentations on the surface. The mean diameter of microparticles is shown in Table 3. The results showed that the addition of rosemary extract, compared to distilled water as a loading material, significantly increased the microparticle diameter for all investigated types. Moreover, the variation in coating material (zein and HPMC) significantly increased the diameter of the microparticles. Consequently, the largest diameter noted in Alg/Z/HPMC microparticles was 1087.37 μm and 1034.12 μm for microparticles with rosemary extract and water, respectively. This is supported by the research from Hosseini et al. who reported an increase in the diameter of zein-electrospun fibers with higher concentrations of encapsulated rosemary essential oil [27].

#### 3.2.2. Morphological Characterization of Microparticles Using SEM

Dried samples were placed on high-conductive graphite tape for SEM analysis. The surface morphology of prepared microparticles was observed using SEM and is presented in Figure 2 at the magnifications of 100×, 5000×, and 20,000×, respectively. The surface of microparticles changed with the composition. The surface of all microparticles was wrinkled and very porous with pores of different sizes. The surface of Alg microparticles was the smoothest with quite a regular structure. The addition of other biopolymers (zein and HPMC) induced significant changes in the surface structure and increased the roughness while reducing the porosity of microparticles. However, the surface of Alg/Z/HPMC microparticles was smoother and with reduced roughness compared to Alg/Z and Alg/HPMC ones. This could be supported by the explanation that the presence of HPMC reduced the agglomeration of zein [18].

The surface imaging results of SEM microscopy were compared to that of microparticles prepared with distilled water instead of the rosemary extract, as presented in Figure 3. Compared to microparticles loaded with distilled water, those with rosemary extract had significant changes in surface morphology. The addition of rosemary extract caused a rougher structure with sharper edges of wrinkles that resemble the furrows of the human intestine. In comparison, a study by Sheng et al. reported that SEM analysis showed a more compact surface of microparticles composed of calcium alginate and HPMC when compared to the blank calcium alginate ones. A compact surface without pores hindered the diffusion during passage through the gastrointestinal tract and additionally slowed the extract release in simulated gastric conditions [21].

SEM microscopy allowed the determination of the surface elemental composition via EDS spectra analysis. The example of a spectrum of Alg/Z/HPMC microparticles is presented in Figure 4. The analysis using EDS, applied to the nearest surface of microparticles, revealed that the major percentage corresponded to carbon and oxygen. The same trend with the dominant peaks was noted in all formulations. Low amounts of sodium and chloride were noted, which are probably residues during microparticle preparation. Similar results were found in the paper by Kolar et al. [45]. The SEM-EDS analysis is a non-destructive analytical technique for the sample, and it gives information on the chemical composition of the surface. The obtained results suggested that rosemary extract is completely encapsulated inside the biopolymer particles, and it is not present on the surface.

#### 3.2.3. Morphology of Microparticles Determined Using AFM

The AFM analysis (Figure 5) of obtained microparticles loaded with rosemary extract was performed to complement SEM surface morphology data. The scanned sample area is presented using topographic images of height data presented as the “top view” and 3D surface view, with appropriate color scale characterizing the microparticle height. All AFM imaging was performed at different regions of each microparticle to ensure the consistency of the obtained results.

The AFM analysis was presented as 3D-topographic height images and amplitude images data in Figure 5. The surfaces of microparticles showed substructures consisting of abundant smaller grains, but in comparison with Alg/Z/HPMC microparticles with sharp boundaries, grains are oriented in different ways in the remaining samples with increased alginate percentage. The average size and mean diameter of grains on the surface are presented in Table 4. The individual grains on the surface area are visible with a height of around 8 nm and a diameter that ranged from 25 ± 64 (for Alg microparticles) to 77 ± 59 nm (for Alg/HPMC microparticles). The 3D topographic images of microparticles showed the finer grain morphological characteristics of microparticles prepared only with calcium alginate (Alg). This is followed by the results of grain diameter. Similar results were found in the work by Jurić et al. who investigated the morphology of alginate microparticles [46].

The roughness analysis of a single microparticle was performed using AFM, and the results are presented in Table 4. The surfaces of the investigated microparticles were highly rough (from Ra = 28 ± 2 to Ra = 46 ± 3 nm for Alg/Z/HPMC and Alg/HPMC microparticles, respectively) and changed with the composition of microparticles. The surface of the microparticles is very rough and heterogeneous, with a lot of small grains and some bigger grains. For that reason, great deviations between grains are notable. The limitation of the instrument is that it provides insight into only a small part of the microparticle surface. In the analyzed surface roughness, the results showed that there was no statistical difference in grain height and grain diameter of microparticles due to variations in grains. Average roughness (R_a_) and root mean square of roughness (R_q_) showed that there was no statistical difference among Alg, Alg/Z, and Alg/Z/HPMC microparticles; however, Alg/HPMC microparticles were showed to have statistically higher roughness. The Z range was the highest for Alg/Z/HPMC microparticles (432 ± 21 nm).

### 3.3. Functional Properties of Microparticles Composed of Variable Coating Materials

In the third section, the functional properties of obtained microparticles with rosemary extract are described. The analysis included the determination of encapsulation efficiency (EE), loading capacity (LC), and TP (Table 5). A cumulative release of polyphenols from microparticles was also observed. Additionally, molecular interactions in microparticles were investigated via FTIR analysis.

#### 3.3.1. Encapsulation Efficiency (EE), Loading Capacity (LC), and Total Protein Content (TP)

To determine the content of polyphenols in obtained microparticles and to assess the efficiency of the encapsulation process, EE and LC were determined. Results are shown in Table 5. High results of EE were noted for all microparticles (113.31 to 120.59%) with approximately similar effectiveness. The highest encapsulation efficiency was measured for Alg/Z/HPMC microparticles, which were the only ones with statistically significantly higher EE compared to all other microparticles. Efficiencies greater than 100% are attributed to the fact that the rosemary extract was added both to the encapsulation solution (biopolymers) and the CaCl_2_ solution. Due to this encapsulation step, no losses of polyphenols via diffusion from microparticles to CaCl_2_ solution occurred [45,47].

The results of loading capacity showed that the lowest LC was noted in microparticles prepared with only calcium alginate (Alg) with the value of 5.55 mg GAE/g, while the introduction of co-biopolymers significantly increased the LC. The LC ranged with increasing results as follows: Alg < Alg/HPMC < Alg/Z/HPMC < Alg/Z. Furthermore, the LC increased in accordance with TP (Table 5). This could be the result of possible interactions between proteins and phenolic compounds [48]. The highest TP was noted in microparticles composed partly of zein, which is expected since zein is a protein itself. These results are following the work of Papoutsis et al. who compared the LC of capsules with and without protein (soy protein). Their results also showed that particles containing the protein had significantly higher LC compared to non-protein freeze-dried particles [49]. The LC and TP are also highly correlated with the results of AFM and SEM. The microparticles with lower LC and TP had rougher surfaces.

#### 3.3.2. Molecular Interactions in Microparticles Formulations

Designing and preparing microparticle formulations for a specific purpose requires a good knowledge of the molecular interactions between biopolymers and the encapsulated material. In Figure 6, the FTIR spectra of individual biopolymers (zein and HPMC) as well as rosemary extract are shown. The characteristic spectra of pure sodium alginate and CaCl_2_ have been reported earlier [36,45]. Characteristic FTIR bands of both constituents are in accordance with literature data [50,51]. Characteristic FTIR bands of zein and HPMC with assignments are listed in Table 6 and are following literature data [52,53,54]. For zein spectra, the presence of a higher amount of α-helices secondary structure is confirmed by the symmetrical spectral peak at 1643 cm^−1^ [54,55,56]. The FTIR spectra of rosemary extract exhibited two characteristic peaks. One broad absorption in the wavelength range (3376–3241 cm^−1^)—corresponding to OH stretching bands of alcohols and/or carboxylic acids vibrations and one at 1627 cm^−1^, which corresponds to bending out of plane C-H and stretching vibrations of conjugated C-C of a benzenoid ring, respectively, in the rosemary extract [57].

On the comparison of FTIR spectra of pure constituents (Figure 6) and dry formulations of Alg microparticles (Figure 7a), changes in the position and intensity of individual bands are noticeable. This is especially noticeable with a decrease in the intensity and position of the band corresponding to the elongation of the carboxyl group (1556 cm^−1^), but also with the band corresponding to the elongation vibration region; the O-H bond in calcium alginate is narrower than the absorption region in sodium alginate. The difference occurs due to the participation of hydroxyl and carboxylate groups of sodium alginate with calcium ions in the process of forming a chelate structure, which leads to the process of reducing hydrogen bonds between hydroxyl functional groups. The vibrations of the asymmetric elongation of the carboxylate ion are shifted towards lower values of the wavenumbers because the calcium ion replaces the sodium ion in the sodium alginate, and there is a change in the charge density, radius, and atomic weight of the cation. These carboxylate group-linked bands are extremely useful for monitoring changes in alginate structure. The obtained results are following the literature data [58]. Compared to microparticles prepared with distilled water, the presence of rosemary extract in the Alg formulation causes slight changes in the intensity of individual bands (carboxyl group 1556 cm^−1^) and with the band corresponding to the elongation vibration region of the O-H bond (wavelength range (3376–3241 cm^−1^). These observations represent evidence of an effective extract encapsulation [59].

The FTIR spectroscopy studies reveal that molecular interaction of zein and alginate occurs in the microparticles (Figure 7b). The spectrum of alginate shows bands at 3337, 1609, 1414, and 1034 cm^−1^ that can be attributed to (H_2_O and OH), (-COO), and (C-O-C) vibrations, respectively. The cross-linking process with Ca^2+^ causes an obvious shift of the 1609 and 1414 cm^−1^ bands to higher wavenumber values, while the band at 1034 cm^−1^ appears as in the polysaccharide, indicating the formation of an ionic bond between calcium ions and deprotonated carboxylate groups of alginate. Furthermore, the FTIR spectrum of the zein is characterized by the presence of a band at 3308 cm^−1^, which is assigned to the NH stretching vibration mode of the so-called amide A groups from the protein. Bands at 1658 and 1538 cm^−1^ are attributed to vibrations of C=O of amide I bond C-N-H of amide II of peptide groups, respectively. The amide I band is essentially associated with the stretching vibration mode of the carbonyl group (C=O), although it also receives a contribution of the C-N stretching and the C-C-N deformation vibrations. However, the amide II band is mainly due to the NH bending vibration mode. It is a mixed contribution of the N-H in-plane bending, the C-N stretching, and the C-C stretching vibrations. In the FTIR spectrum of the Alg/Z microparticles, the band at 1658 cm^−1^ characteristic of amide I vibrations of zein appears to be overlapped by the alginate characteristic band at 1609 cm^−1^, being now observed only as a single band at 1614 cm^−1^. This band is asymmetric, suggesting that there is a band at a low frequency related to the amide I component and a high-frequency band associated with the amide II component, which is an indication that a hydrogen-bonded aggregate is formed. This effect can be a consequence of possible interactions between the carboxylate groups in alginate with protonated amino groups from the protein. The band corresponding to vibrations of the amide II groups of zein appears as a low-intensity band, being displaced at 1545 cm^−1^. These shifts suggest the existence of interactions between the protein and the polysaccharide. In the FTIR spectrum of the Alg/Z microparticles after cross-linking with Ca^2+^ ions, a shift of the (COO) band to 1632 cm^−1^ is observed, in a similar way to that observed in cross-linked alginate. However, the band ascribed to OH vibration modes of alginate appears at a higher wavenumber (3462 cm^−1^) than in cross-linked alginate. This phenomenon suggests the existence of hydrogen bonding interactions between the biopolymers, similar to that reported for xanthan–alginate blends [60]. In comparison to microparticles prepared with distilled water, the presence of aqueous rosemary extract in the Alg/Z microparticles causes intensive changes in the intensity and position of individual bands. The spectrum of the Alg/Z microparticles prepared with rosemary extracts incorporated in the matrix shows a displacement of the typical COO bands appearing at 1609 cm^−1^ in the pure formulations and 1596 cm^−1^ formulations with extracts. Moreover, the change of bend in the range of 3359–3207 cm^−1^ indicates the change in hydrogen bonding because of the presence of rosemary extracts. The characteristic bands of rosemary extracts are overlapped with the bands of biopolymers [60]. Similar results were found in a paper from Hosseini et al. who performed encapsulation of rosemary essential oil in zein via the electrospinning technique [27]. A higher wavenumber was observed as a result of adding rosemary essential oil to zein, indicating a lower α-helix length. The rosemary essential oil affected the secondary structure of zein protein and confirmed that the rosemary essential oil was successfully encapsulated.

The FTIR spectra of Alg/HPMC formulations (Figure 7c) with and without the rosemary extract are very similar with changes in peak intensity when rosehip extracts are encapsulated. In both cases, a broad peak at the intensity of 3341 cm^−1^ was observed, which indicates the presence of stretching of the hydroxyl groups of HPMC, but also the presence of hydrogen bonds, which are strengthened by the addition of rosemary extract. There is also an increase in the intensity of the peak at the value of 1021 cm^−1^, which corresponds to the process of cross-linking of calcium ions with sodium alginate, i.e., electrostatic interaction between calcium ions and the hydroxide group of sodium alginate [61,62].

The addition of both zein and HPMC into calcium alginate-based formulations caused some changes in the spectra of formulations samples (Figure 7d). The peaks at 2926, 1696, and 1461 cm^−1^ were shifted to higher wavenumbers. These spectral changes can be attributed to the possible interaction (hydrogen bonds) between HPMC, calcium alginate, and zein. It can be observed that hydrogen bonding in COO groups was checked in Alg/Z/HPMC formulations by shifting the peak at 1647 cm^−1^ for COO- asymmetric stretching and shifting the peak at 1455 cm^−1^ for COO- symmetric stretching [63]. The presence of rosemary extract in the Alg/Z/HPMC formulation causes intensive changes in the intensity and position of individual bands.

#### 3.3.3. In Vitro Release Profiles of Total Polyphenols from Microparticle Formulations

The antioxidant potential of polyphenols, i.e., their potential bioactivity in vivo depends on their metabolism, absorption, distribution, and excretion from the body after their intake and the reducing properties of the resulting metabolites [64]. As TF, DPPH, and ABTS have high correlations with TPC, only TPC was chosen as the most propriate method for the evaluation of bioactivity of valuable phenolic compounds from rosemary [10,65]. For that reason, the in vitro TPC release from microparticles was observed in neutral (distilled water), acidic HCl solution (gastric simulation), and alkaline buffer solution (intestinal simulation) at 37 °C. The results are shown in Figure 8 as cumulative release presented as a percentage (%) of the total LC of each prepared microparticle type over time.

The TPC release in pure water showed quick initial release (11.6–40.6% in the first 15 min) that was followed by slower gradual release (Figure 8). The addition of HPMC in the initial formulation (Alg/HPMC microparticles) increased cumulative release compared to Alg microparticles by 19% on average. On the other side, zein significantly decreased the cumulative release of TPC (Alg/Z microparticles), by an average of 67%. Therefore, zein causes a slowdown in the polyphenols release in water. The isoelectric point of zein protein is at pH 6.2, and for that reason, it is poorly soluble in neutral pH. Microparticles with both HPMC and zein (Alg/Z/HPMC) had on average 26.7% lower release in time compared to Alg microparticles. The cumulative release trend was in between the trend of Alg/Z and Alg microparticles. The results of cumulative release in water are correlated with microparticles morphology analyzed using SEM and AFM. The microparticles with rougher surface and higher grains on the surface had less pores and slower release of TPC.

In vitro simulated gastric TPC release analyzed in HCl solution (pH 1.64) showed changes in release trend compared to release in water. In an acidic medium, Alg microparticles had increased cumulative release in time compared to a neutral medium (distilled water). With this faster release, after 120 min, 27.3% more polyphenols were released in the tested medium, and the highest proportion of released polyphenols from all four formulations, during 120 min. The release rate of polyphenols from microparticles in an acidic medium was in the following order: Alg > Alg/HPMC > Alg/Z/HPMC > Alg/Z. The addition of both HPMC and zein into the microparticle structure decreased the TPC release in HCl. Consequently, Alg/Z microparticles had the lowest TPC release during time, which was 1.38 × lower after 120 min compared to release in water. According to these results, it is evident that compared to the microparticles without zein in their structure, the microparticles with zein release polyphenols in HCl more slowly. These results concur with the results of the work of Karthikeyan et al. where they compared the release from microparticles coated with zein and microparticles without zein. The addition of zein to the microparticle structure slowed down the release in the acidic medium [66].

Furthermore, the cumulative release was analyzed in a buffer solution (pH 7.40) for simulated intestinal conditions. When compared to water and HCl, the release drastically changed. The greatest increase in the concentration of released polyphenols takes place in the first 15 min, that is when the release is the fastest. In that period, the highest proportion of released polyphenols was measured in microparticles containing calcium alginate, HPMC, and zein (Alg/Z/HPMC—43.8%). After this period, the rate of release of polyphenols from this type of microparticles decreases, and after two hours, the measured proportion of released polyphenols is 77.2%—the smallest proportion of polyphenols released from microparticles in phosphate buffer. No polyphenols were released in the first minute from any microparticle formulations. After that, the rapid growth of release rate occurs, and after two hours, the complete release of polyphenols from the microparticles (100%) was measured in both Alg and Alg/Z microparticles. From the measurement results, it is evident that in phosphate buffer, microparticles containing HPMC release polyphenols more slowly than microparticles that do not contain HPMC. The results are consistent with the results obtained in the work of Patole and Pandit (2018) where the release of mesalamine from capsules coated with HPMC slows down the release under small intestine simulation conditions [67]. The total release of TPC in phosphate buffer happened due to the rapid degradation of microparticles. The degradation is a consequence of ion exchange between sodium and calcium cations that causes a gel network to relax, diffusion of calcium ions from the gel to the medium, and finally microparticle erosion [45,68].

Comparing the results of measuring the release of polyphenolic compounds in water, HCl, and phosphate buffer, it is evident that each type of microparticle reacts differently relative to the medium type. All types of microparticles release polyphenols the fastest in phosphate buffer, while it was the slowest in acidic conditions (except for Alg microparticles that had the lowest release rate in water). Therefore, it is evident that the addition of co-biopolymers (HPMC and zein) to the microparticle composition slowed down the release of polyphenols from the particles (except for Alg/HPMC in water). The TPC release rate increased with high pH for microparticles composed of alginate, zein, and/or HPMC (Alg/Z, ALG/HPMC, and Alg/Z/HPMC). The pH significantly affected the release of rosemary polyphenols from microparticles as a result of zein’s high level of solubility in acidic media. This was also confirmed by Hosseini et al. who analyzed the release of rosemary essential oil from zein-electrospun fibers [27] and by Colín-Orozco et al. who reported higher rates of release with a pH increase of rosemary extract from polyethylene oxide/whey protein isolate fibers [69]. The release level of grape seed proanthocyanidin extract from microparticles prepared with calcium alginate and HPMC also showed an acceleration of release and microparticles disintegration with higher pH in work from Sheng et al. [21]. The solubility is also correlated with possible protein-phenols interactions that showed higher solubility in alkaline pH, and therefore the release of phenolic compounds [48].

Thus, most of the polyphenols will remain in the microparticles after passing through the stomach and will be released from the microparticles in the small intestine. This is important since most polyphenols are metabolized in the intestine by the microflora (mostly in the jejunum and ileum) [64]. With the right combinations of biopolymers, the desired level of phenolic release and stability can be achieved, depending on the desired purpose. The investigated formulations of microparticles show great potential in the production of functional foods, where different polymer combinations can be used depending on the desired food product properties.

The potential limitations of our study include more precise simulation of gastrointestinal tract conditions that can include enzymatic digestion, and application in real food systems. Microparticle composition is important to consider when implementing the latter into food production. Future studies should focus on the investigation of microparticle behavior in real, complex matrices, such as food systems where interactions of many components might interfere with the active ingredient release.

## 4. Conclusions

In this research, rosemary extract obtained using HVED has proven to be an effective potential source of bioactive compounds, especially polyphenols. The microencapsulation of aqueous rosemary extract was successfully carried out by the ionic gelation method using calcium alginate, zein, and HPMC as coating materials. Each formulation resulted in specific microparticles different in physicochemical characteristics, and functional properties in terms of bioactive content and release properties.

The swelling behavior significantly changed (*p* < 0.05) with the addition of HPMC to calcium alginate, while zein and rosemary extract did not make significant changes to the swelling degree of microparticles. The mean diameter of dry microparticles increased with the addition of rosemary extract and with both zein and HPMC to calcium alginate. The morphological surface of microparticles analyzed using SEM and AFM was shown to be granular with visible individual grains of 6.3–11.3 nm in height. The addition of the second biopolymers to calcium alginate increased the roughness and reduced the porosity of microparticles. Moreover, the addition of rosemary extract caused a rougher structure with sharper edges of grains on the surface of the microparticles.

The encapsulation efficiency for all formulations was high (>100%), indicating an efficient encapsulation process where rosemary extract was added to both encapsulation solution (biopolymers) and CaCl_2_ solutions before encapsulation. The loading capacity of polyphenols from rosemary extract was the highest in microparticles prepared with calcium alginate and zein (10.42 ± 0.72 mg GAE/g). Furthermore, the FTIR analysis with and without rosemary polyphenols presented specific molecular interactions for each formulation. Blending alginate with other polymers influenced molecular interactions, mainly hydrogen bonds and electrostatic interactions. A better understanding of molecular interactions between microparticle constituents ensured its potential for controlling their release behavior and the further development of new microparticles for specific use. Microencapsulation provided a protective effect for rosemary polyphenols against pH changes during digestion. The addition of both HPMC and zein decreased the polyphenols release in gastric conditions, which resulted in an increase of polyphenol bioavailability in the gut. The results indicated a high potential of microparticles loaded with rosemary extract for further use, especially in functional food development.

## Figures and Tables

**Figure 1 foods-12-01570-f001:**
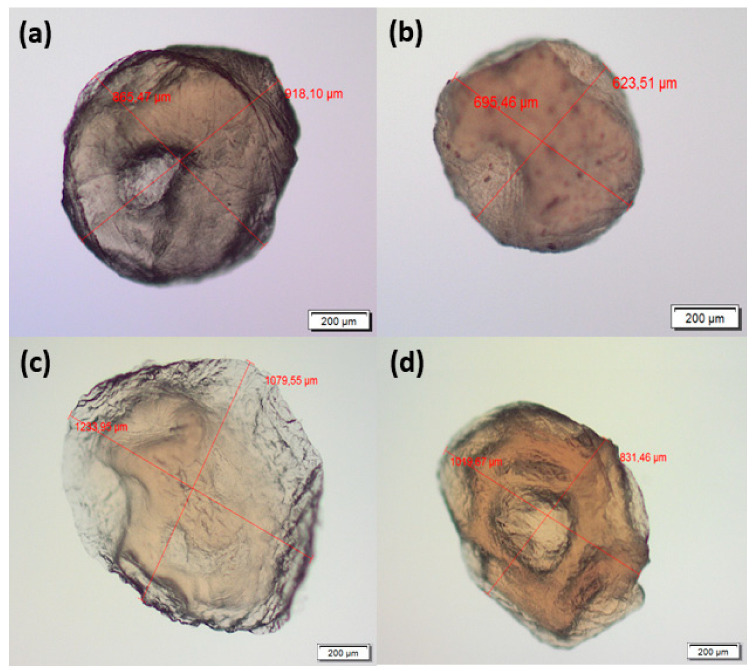
Images of dry microparticles obtained under an optical microscope (Leica MZ16a stereomicroscope, Leica Microsystems Ltd., Switzerland): (**a**) Alg (**b**) Alg/Z (**c**) Alg/HPMC, and (**d**) Alg/Z/HPMC. Bars are indicated.

**Figure 2 foods-12-01570-f002:**
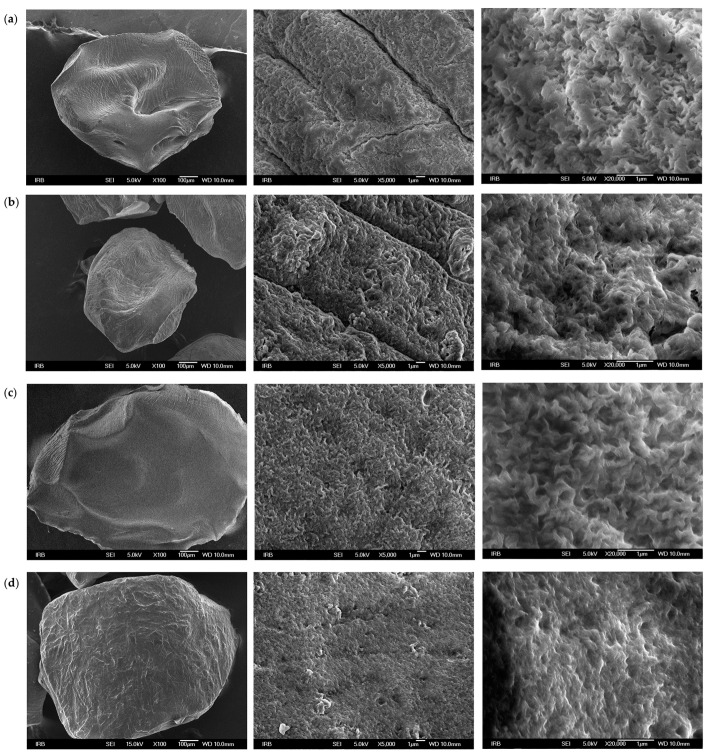
SEM microphotographs of (**a**) Alg (**b**) Alg/Z (**c**) Alg/HPMC, and (**d**) Alg/Z/HPMC microparticles loaded with rosemary extract. Bars are indicated.

**Figure 3 foods-12-01570-f003:**
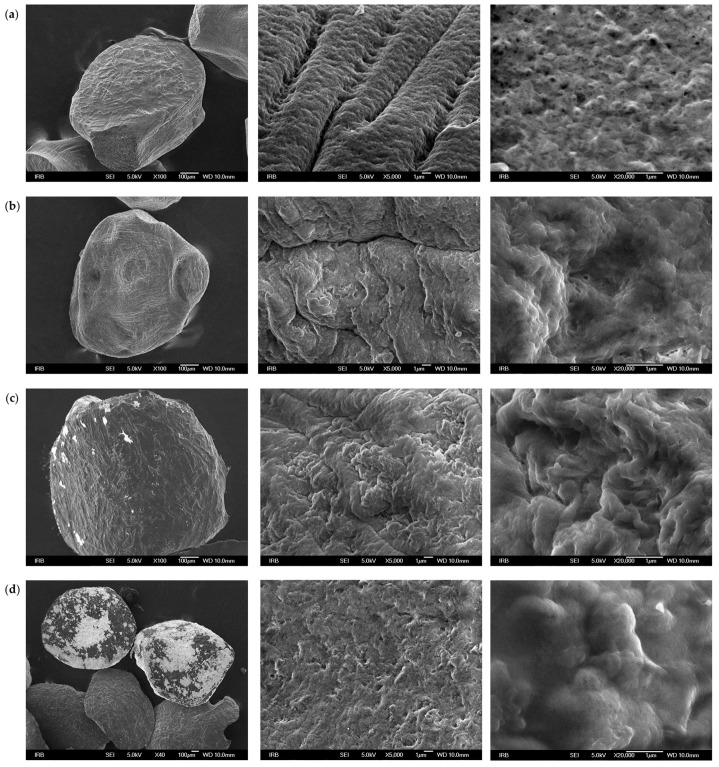
SEM microphotographs of (**a**) Alg (**b**) Alg/Z (**c**) Alg/HPMC, and (**d**) Alg/Z/HPMC microparticles loaded with distilled water. Bars are indicated.

**Figure 4 foods-12-01570-f004:**
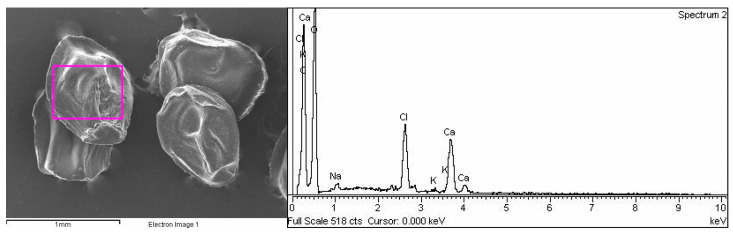
Morphology and surface elemental analysis using EDS (expressed in the atomic weight percent) of Alg/Z/HPMC microparticles loaded with rosemary extract. Bar is indicated.

**Figure 5 foods-12-01570-f005:**
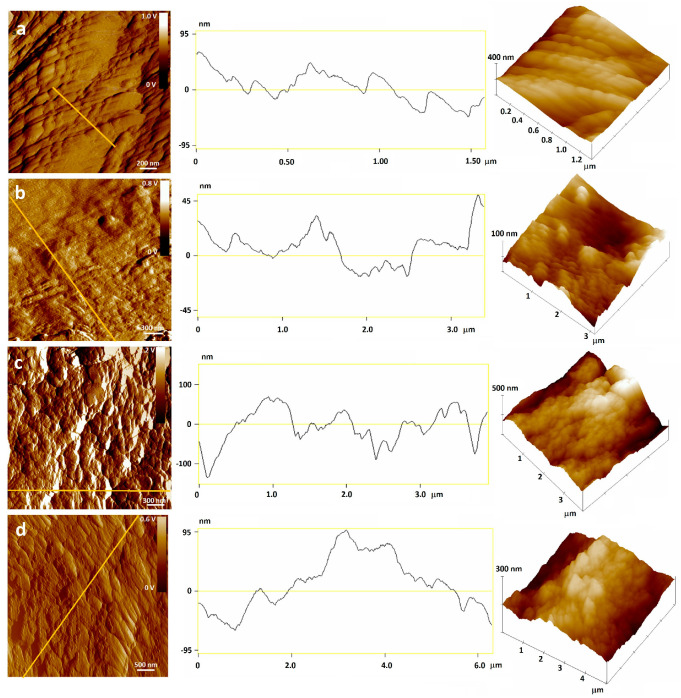
AFM amplitude image (**left**), section analysis profile along the labeled line (**middle**) and 3D-topographic images of height data—top view (**right**) of microparticles loaded with rosemary extract: (**a**) Alg, (**b**) Alg/Z, (**c**) Alg/HPMC, and (**d**) Alg/Z/HPMC. Bars are indicated.

**Figure 6 foods-12-01570-f006:**
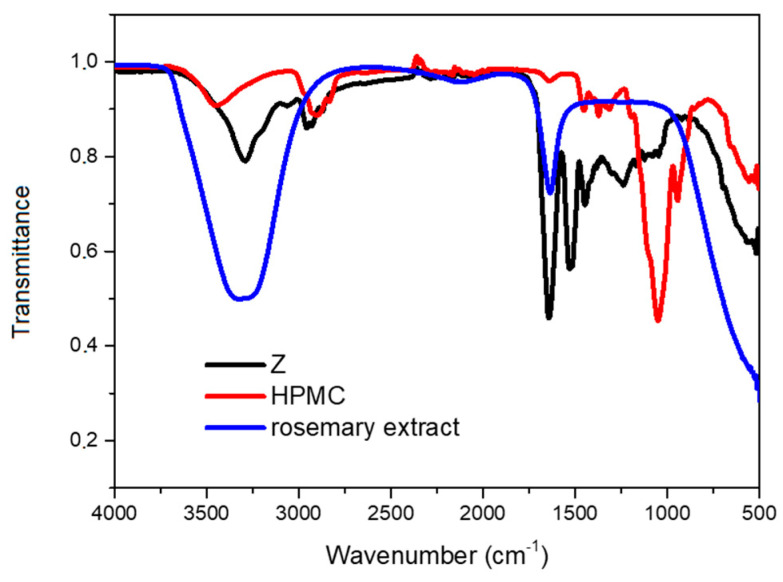
FTIR spectrum of zein (Z)-black line; HPMC—red line; and rosemary extract—blue line.

**Figure 7 foods-12-01570-f007:**
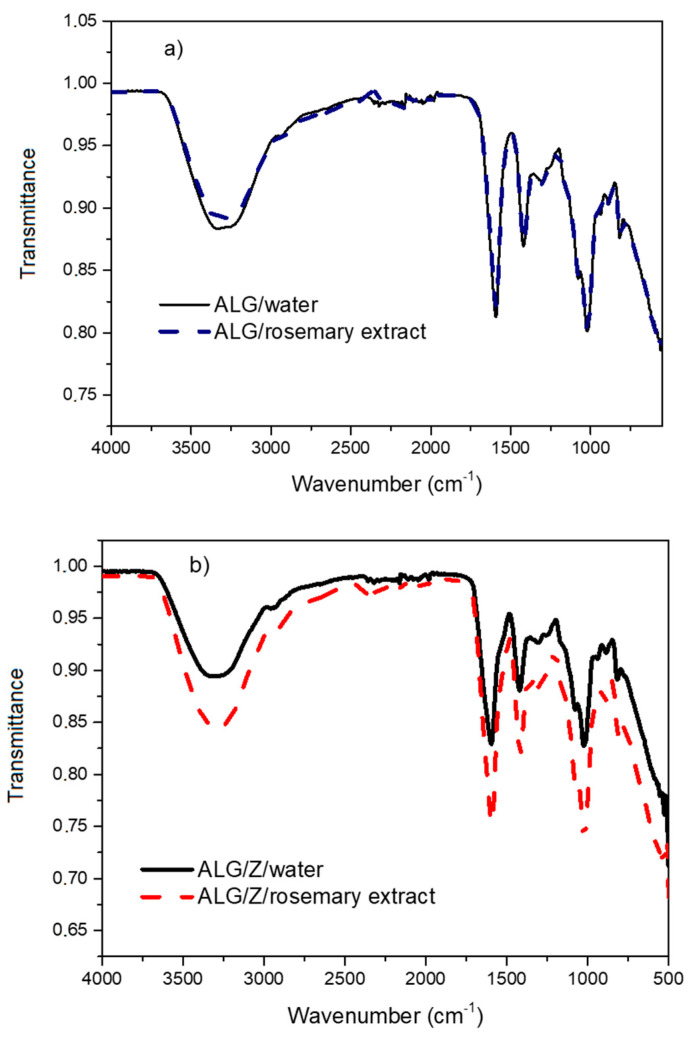
FTIR spectra of microparticles loaded with distilled water and rosemary extracts: (**a**) ALG/water—black line, ALG/rosemary extract—blue line; (**b**) ALG/Z/water—black line, ALG/Z/rosemary extract—red line; (**c**) ALG/HPMC/water—black line, ALG/HPMC/rosemary extract; (**d**) ALG/ZHPMC/rosemary extract—green line.

**Figure 8 foods-12-01570-f008:**
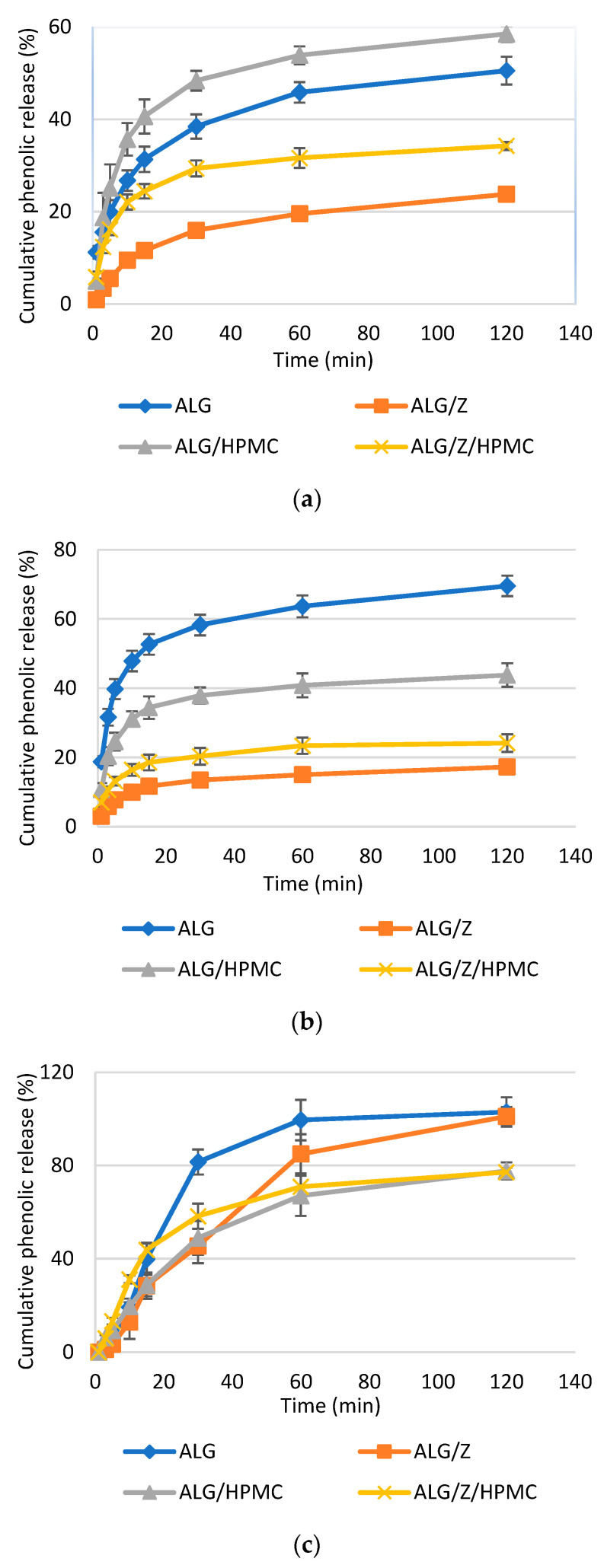
Cumulative release of total polyphenolic content (TPC) from microparticles in different dissolution media: (**a**) distilled water; (**b**) HCl solution (pH 1.64); and (**c**) buffer solution (pH 7.40). The error bars indicate the standard deviation of the means.

**Table 1 foods-12-01570-t001:** Encapsulation parameters.

Microparticles	Vibration Frequency (Hz)	Amplitude	Pressure (mbar)
Alg	50	3	30
Alg/Z	60	3	20
Alg/HPMC	40	3	80
Alg/Z/HPMC	40	3	60

**Table 2 foods-12-01570-t002:** Chemical properties of rosemary extract (the total polyphenolic content (TPC), total flavonoid content (TF), antioxidant capacity (ABTS and DPPH methods), and total protein content (TP)).

Method	Determination Value
TPC (mg GAE/L)	333.07 ± 12.32
TF (mg QE/L)	333.08 ± 24.22
ABTS (mmol TE/L)	1.94 ± 0.15
DPPH (mmol TE/L)	1.98 ± 0.16
TP (mg BSA/mL)	8.48 ± 0.07

Results are expressed as mean ± standard deviation.

**Table 3 foods-12-01570-t003:** Physical properties of microparticles with different coating materials loaded with rosemary extract and distilled water as loading materials (swelling degree and diameter of microparticles).

Microparticles	Loaded Material	Swelling Degree (%)	Diameter (μm)
Alg	water	55.88 ± 1.79 ^bA^	651.29 ± 79.22 ^aA^
	extract	52.63 ± 6.98 ^aA^	698.79 ± 90.39 ^aB^
Alg/Z	water	48.95 ± 1.82 ^aA^	691.32 ± 60.17 ^bA^
	extract	50.57 ± 5.86 ^aA^	711.57 ± 69.12 ^bB^
Alg/HPMC	water	126.75 ± 11.64 ^cA^	874.51 ± 166.64 ^cA^
	extract	114.72 ± 22.97 ^bA^	1202.90 ± 311.42 ^cB^
Alg/Z/HPMC	water	138.95 ± 12.34 ^cA^	1034.12 ± 163.56 ^dA^
	extract	119.83 ± 11.94 ^bA^	1087.37 ± 236.51 ^dB^

Results are expressed as mean ± standard deviation; values presented with the different lower case are statistically different at *p* = 0.05 for different microparticles (coating material) but the same loading material; values presented with the different upper case are statistically different at *p* = 0.05 for different loading material but same coating material.

**Table 4 foods-12-01570-t004:** Roughness parameters of the microparticle surfaces (grain mean height, grain mean diameter, average roughness—R_a_, root mean square of roughness—R_q_ and Z range for microparticles loaded with rosemary extract).

Microparticles	Grain Height (nm)	Grain Diameter (nm)	R_a_ (nm)	R_q_ (nm)	Z (nm)
Alg	7.9 ± 5.5 ^a^	25 ± 64 ^a^	31 ± 2 ^a^	39 ± 3 ^a^	327 ± 22 ^a^
Alg/Z	8.5 ± 4.2 ^a^	76 ± 164 ^a^	29 ± 1 ^a^	38 ± 2 ^a^	342 ± 16 ^a^
Alg/HPMC	11.3 ± 2.8 ^a^	77 ± 59 ^a^	46 ± 3 ^b^	53 ± 4 ^b^	340 ± 13 ^a^
Alg/Z/HPMC	6.3 ± 5.5 ^a^	58 ± 54 ^a^	28 ± 2 ^a^	38 ± 1 ^a^	432 ± 21 ^b^

Results are expressed as mean ± standard deviation; values presented with different letters are statistically different at *p* = 0.05.

**Table 5 foods-12-01570-t005:** Functional properties of microparticles with different coating materials (encapsulation efficiency (EE), loading capacity (LC), and total protein content (TP)).

Microparticles	EE (%)	LC (mg GAE/g)	TP (mg BSA/mL)
Alg	110.94 ± 2.14 ^a^	5.55 ± 0.42 ^a^	3.73 ± 0.68 ^a^
Alg/Z	117.75 ± 7.28 ^a^	10.42 ± 0.72 ^d^	11.31 ± 1.47 ^d^
Alg/HPMC	113.31 ± 3.26 ^a^	6.74 ± 0.74 ^b^	4.30 ± 0.81 ^b^
Alg/Z/HPMC	120.59 ± 2.37 ^b^	9.33 ± 0.62 ^c^	9.77 ± 1.29 ^c^

Results are expressed as mean ± standard deviation; values presented with different letters are statistically different at *p* = 0.05.

**Table 6 foods-12-01570-t006:** FTIR bands of zein and HPMC with assignments.

Coating Material	Vibration (cm^−1^)	Assignment
Zein	1643	Amide I (C=O stretching)
	1531	Amide II (N-H bending)
	1240	Amide III (C-N stretching)
	3292	N-H stretching
	3304	O-H stretching
	2958	reflected the stretching vibration of the intermolecular bonded hydroxyl group
HPMC	3444	stretching frequency of the hydroxyl (-OH) group
	1373	bending vibration of -OH
	2929, 1055	stretching vibration bands related to C-H and C-O
	1300 to 900	C-O-C stretching
	3000–2800	C-H stretching

## Data Availability

Data are contained within the article.

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
