# Peer review of "Encapsulation of Rosemary Extracts using High Voltage Electrical Discharge in Calcium Alginate/Zein/Hydroxypropyl Methylcellulose Microparticles"

_foods, 2023, doi:10.3390/foods12081570_

Round 1

Reviewer 1 Report

The current manuscript is well-written and the proposed contents are relevant. The objective is clearly defined and the result tasks are correctly formulated. Therefore, some comments:

1- Line 119, 131, 154, 161, 356, …. etc.:  The reference should not be in italics and without the year. Please check all and correct them.

2- Line 173-174, 181 and Table 2: Abbreviation BSA, TPC and TE: please write the full name in the first instance and follow it immediately by the abbreviated version in brackets.

3- As you are using the abbreviation “TPC” for “total polyphenols content”, why not use the abbreviation also for total flavonoids and total protein contents?

4- Line 296: remove the word “Figure 5”, it is written before in line 295.

5- Line 365: reference citation in the text is missing.

Reviewer 2 Report

I have the following concerns after carefully reading the article:

1 The authors declaimed the HVED is novel, so please add their advantages in the INTRDUCTION. (Line 48)

2 Line 103, the plant materials were harvested in 2017, how can we ensure the quality has been maintained?

3 Line 138, the solution's heating temperature and pH are unclear.

4 Line 205, before the FT-IR determination, were the samples dried?

5 Table 2, the “extract” should be the determination value.

6 Table 4, for the sample Alg/Z, the value of grain diameter was probably wrong. Furthermore, these data should be included in the statistical analysis.

7 The wavelength in Figures 6 and 7 should be labeled in order to reflect the changes in molecules. How did the authors measure the FT-IR spectrum when water was loaded into the sample in Figure 7?

8 Figure 8, please remove the background line.

9 Lines 393/664, here “α<0.05” should be “P<0.05”.

10 Line 414, SEM/AFM results should be correlated with microparticle functional properties

11 Lines 598/602, the “minutes”, should be “min”

12 Please include the FT-IR results in the CONCLUSION.

13 The format is uniform, but there are many errors in the references.

Reviewer 3 Report

The article titled " High voltage electrical discharge rosemary extracts encapsulation in calcium alginate/zein/hydroxypropyl methylcellulose microparticles” by Nutrizio et al., characterizes different microparticles encapsulated with rosemary extracts. The current article lacks clarity and misses some crucial details, which makes it hard to interpret the following datasets. The manuscript could be reconsidered after the following queries are addressed. 

1.     The results and discussion are separate sections which makes it difficult to correlate the discussion with the data showed in the earlier part of the document. It would be helpful to keep the discussion after respective results for better understanding.

2.     The loading capacity formula (line 184) shows twice consideration for TPCtot

3.     The authors investigate the different chemical properties of rosemary extract, however fails to conclude the importance and the effect of each or the predominant component which could be essential in functional food development

4.     The EDS analysis doesn’t give any essential information other than the presence of carbon and oxygen which is inherent for any polymer.

5.     The FTIR spectra (Fig 6 and 7) are poorly represented. There is spelling mistake in Y axis label. Suggested to have separate spectra with proper labelling of peaks for clarity.

6.     The AFM data is poor in quality and lacks clarity. The morphology is not clear for microparticles and only indicates possible presence of high aggregates.

Reviewer 4 Report

In this study authors have encapsulated the rosemary extract obtained by HVED in biopolyme based microparticles prepared from different combinations of calcium alginate, zein, and HPMC. The obtained microparticles were further evaluated for physical properties. Chemical analysis of the composition and in vitro release profiles of total polyphenols in simulated gastrointestinal tract conditions were also investigated.

The specific recommendations are as follows:

A comparative chemical properties of rosemary extract with or without encapsulation should be performed. 

Although authors used HVED extraction method to obtain rosemary extract, the study does not provide direct novelty. Previously many articles were available on encapsulated rosemary extract along with chemical and biological property. 

Acknowledge the limitations of your study.

-------------------------

Round 2

Reviewer 1 Report

The manuscript is better after major revisions. Therefore, as I recommended in the first report, the references cited in-text should be without the year, please remove it. Example: line 122: should be “Nutrizio et al.” instead of “Nutrizio et al. (2020) “.

Author Response

Thank you for your comment and we apologize for misunderstanding and not removing the years in the first review. We have removed years from the references throughout the whole manuscript.

Reviewer 3 Report

The authors have addressed the comments. Minor change suggested as follows:

Scale bars of SEM images (Fig 2 and 3) not properly visible. Suggested to correct that. 

Author Response

Thank you for your suggestion. Figures 2 and 3 have been moved to separate pages at the biggest size possible, for easier reading of scales.  As requested by the editorial board, each SEM figure (32 in total) was sent separately and will be processed in its original size and picture quality upon publishing.

Reviewer 4 Report

NA

Author Response

Thank you for your review. The English language was revised throughout the whole manuscript.